# Nano-Graphene Oxide-Promoted Epithelial–Mesenchymal Transition of Human Retinal Pigment Epithelial Cells through Regulation of Phospholipase D Signaling

**DOI:** 10.3390/nano11102546

**Published:** 2021-09-28

**Authors:** Sun Young Park, Woo Chang Song, Beomjin Kim, Jin-Woo Oh, Geuntae Park

**Affiliations:** 1Bio-IT Fusion Technology Research Institute, Pusan National University, Busan 46241, Korea; 201210503@pusan.ac.kr; 2Department of Nanofusion Technology, Pusan National University, Busan 46241, Korea; dck3202@naver.com (W.C.S.); ojw@pusan.ac.kr (J.-W.O.)

**Keywords:** nano-graphene oxide, retinal pigment epithelium, cell migration, epithelial-to-mesenchymal transition, phospholipase D

## Abstract

Nano-graphene oxide (Nano-GO) is an extensively studied multifunctional carbon nanomaterial with attractive applications in biomedicine and biotechnology. However, few studies have been conducted to assess the epithelial-to-mesenchymal transition (EMT) in the retinal pigment epithelium (RPE). We aimed to determine whether Nano-GO induces EMT by regulating phospholipase D (PLD) signaling in human RPE (ARPE-19) cells. The physicochemical characterization of Nano-GO was performed using a Zetasizer, X-ray diffraction, Fourier-transform infrared spectroscopy, and transmission electron microscopy. RPE cell viability assays were performed, and the migratory effects of RPE cells were evaluated. RPE cell collagen gel contraction was also determined. Intracellular reactive oxygen species (ROS) levels were determined by fluorescence microscopy and flow cytometry. Immunofluorescence staining and western blot analysis were used to detect EMT-related protein expression. Phospholipase D (PLD) enzymatic activities were also measured. Nano-GO significantly enhanced the scratch-healing ability of RPE cells, indicating that the RPE cell migration ability was increased. Following Nano-GO treatment, the RPE cell penetration of the chamber was significantly promoted, suggesting that the migratory ability was strengthened. We also observed collagen gel contraction and the generation of intracellular ROS in RPE cells. The results showed that Nano-GO induced collagen gel contraction and intracellular ROS production in RPE cells. Moreover, immunofluorescence staining and western blot analysis revealed that Nano-GO significantly regulated key molecules of EMT, including epithelial-cadherin, neural-cadherin, α-smooth muscle actin, vimentin, and matrix metalloproteinases (MMP-2 and MMP-9). Interestingly, Nano-GO-induced RPE cell migration and intracellular ROS production were abrogated in PLD-knockdown RPE cells, indicating that PLD activation played a crucial role in the Nano-GO-induced RPE EMT process. We demonstrate for the first time that Nano-GO promotes RPE cell migration through PLD-mediated ROS production. We provide preliminary evidence to support the hypothesis that Nano-GO has adverse health effects related to RPE damage.

## 1. Introduction

Since the introduction of graphene-based nanomaterials in a two-dimensional carbon sheet in 2004, various applications of graphene oxide (GO) have been anticipated in many scientific fields, such as sensing, electronics, optical energy, biomedicine, and biotechnology [1,2,3]. Nano-graphene oxide (Nano-GO) is a granular material with sizes ranging from 20 to 100 nm. With its unique properties, Nano-GO has revolutionized biomedical applications through targeted drug delivery and disease diagnosis [4,5]. However, Nano-GO has become a topic of growing concern for human health risks [6,7]. Similar to other particulate matter, several studies have shown that GO is harmful to human health [8,9,10]. Some studies have reported that GO induces reactive oxygen species (ROS), which are capable of causing plasma membrane damage, mitochondrial injury, immune responses, and programmed cell death [11,12,13]. It has been reported that GO may lead to pro-inflammation, cell death, thrombus formation, and cancer metastasis, which appears to be induced by increased ROS production [14,15,16]. Furthermore, GO has been demonstrated to exert advantageous and disadvantageous effects on tumor progression. As a potential tumor promoter, GO can promote the metastasis of human lung, breast, prostate, and liver cancer [17,18]. Furthermore, despite recent studies demonstrating that GO treatment upregulates epithelial-to-mesenchymal transition (EMT) in prostate, lung, breast, and liver cancer, no studies have previously proven a link between Nano-GO and EMT in human retinal pigment epithelium (RPE) cells.

RPE is a critical cellular component of retinal function and integrity. RPE performs diverse functions, such as protection of the retina against photo-oxidation, the formation of the outer blood-retinal barrier, transport of nutrients, oxygen, and ions to the retina, and production of a wide range of growth factors and cytokines, phagocytosis, and retinal remodeling [19,20]. In numerous degenerative retinal diseases, RPE dysfunction leads to disrupted polarization and loss of barrier function, causing proliferative vitreoretinopathy, inherited rod-cone degeneration, inherited macular degeneration, and age-related macular degeneration [21,22]. Additionally, RPE dysfunction is known to trigger the EMT process. EMT is related to RPE morphogenesis and degenerative retinal disease, which induces substantial changes in RPE cell morphology, migration, contraction, and related gene expression. EMT facilitates the acquisition of migratory mesenchymal capacities and the loss of epithelial-specific properties and leads to structural and functional injury to the retina [23,24,25]. Ultimately, the EMT process of RPE is a common feature shared by proliferative vitreoretinopathy and age-related macular degeneration pathogenesis. During the EMT process, the expression of intercellular adhesion molecules, including E-cadherin (epithelial-cadherin), is downregulated, while that of mesenchymal markers, including N-cadherin (neural-cadherin), α-smooth muscle actin (α-SMA), and vimentin, is upregulated, which in turn induces the migration abilities of RPE [26,27,28].

Despite several studies on the toxicological mechanisms of GO, there is still a lack of understanding regarding the effects of nano-sized GO treatment on RPE function. However, the effect of Nano-GO on RPE cells was not the only goal of this study. An important goal was to investigate the effects of Nano-GO on EMT and explore the underlying molecular mechanism in induced human RPE cells. Our results provide essential systematic evidence for understanding the biological activity of Nano-GO in human RPE (ARPE-19) cells.

## 2. Materials and Methods

### 2.1. Nano-GO Characterization

The Nano-GO (1 mg/mL) used in this study was purchased from UniNano Tech Co., Ltd. For size and zeta potential determination, Nano-GO was diluted with deionized water and 10 μg/mL was analyzed using a Zetasizer (Malvern Nano ZS90 series, Malvern, UK). For the X-ray diffraction (XRD) pattern, the Nano-GO was dried using a freeze-drying method and the Nano-GO was determined using an X-ray diffractometer (PANalytical, Almelo, The Netherlands). Fourier-transform infrared (FTIR) spectra of Nano-GO were acquired using an FTIR spectrophotometer (Spectrum GX; Perkin Elmer, Inc., Boston, MA, USA). The surface morphology of Nano-GO was examined using a Hitachi H-7600 transmission electron microscope (TEM, Hitachi High-Technologies Corp., Tokyo, Japan).

### 2.2. Human RPE Cell Culture and Treatment

Human RPE (ARPE-19) cells were acquired from American Type Culture Collections (ATCC, Manassas, VA, USA) and cultured in Dulbecco’s modified Eagle’s medium/F12 (DMEM, Invitrogen-Gibco, Carlsbad, CA, USA) supplemented with 10% heat-inactivated fetal bovine serum (FBS) (Invitrogen-Gibco, Carlsbad, CA, USA), 10 units/mL penicillin, and 100 units/mL streptomycin (PS) (Invitrogen-Gibco, Carlsbad, CA, USA) in a humidified atmosphere of 95% air and 5% CO_2_ at 37 °C. All experiments were carried out 24 h after seeding in 48-well plates at a density of 2.0 × 10^4^ RPE cells/well or in 6-well plates at a density of 5.0 × 10^5^ RPE cells/well. RPE cells were pre-treated with various concentrations (0–200 μg/mL) of Nano-GO. An equal volume of vehicles was used as the control.

### 2.3. RPE Cell Viability

RPE cells seeded in 24-well plates at a density of 4.0 × 10^4^ RPE cells/well were treated with various concentrations (0, 10, 20, 40, 80, 100, and 200 μg/mL) of Nano-GO for 24 h, 48 h, and 72 h. RPE cell viability was analyzed using the Cell Counting Kit-8 assay (Sigma-Aldrich, St. Louis, MO, USA). All procedures were conducted according to the manufacturer’s recommendations.

### 2.4. RPE Cell Migration

RPE cell migration was performed using SPLScar Scratchers (24-well lid, SPL Life Sciences, Pocheon, Korea). All procedures were performed according to the manufacturer’s instructions. Straight-line scratches across RPE cells were made using SPLScar Scratchers and then treated with Nano-GO (20 and 40 μg/mL) for 24 h and 48 h, photographed from the straight-line scratch across RPE cells under a stereomicroscope (SMZ800, Nikon Corporation, Tokyo, Japan). Images were captured using an attached IMT i-Solution CAM 3 (IMT iSolution Inc., Vancouver, BC, Canada). The RPE cell migration assay was performed using the CytoSelect 24-Well Cell Migration Assay (Cell Biolabs, San Diego, CA, USA). All procedures were performed according to the manufacturer’s instructions. RPE cells were treated with Nano-GO (20 and 40 mg/mL) for 24 h. Migratory cells were fixed and stained with cell stain solution, and the whole area was photographed (upper chamber) under a stereomicroscope (SMZ800, Nikon Corporation, Tokyo, Japan). Images were captured using an attached IMT i-Solution CAM 3 (IMT iSolution Inc., Vancouver, BC, Canada). Subsequently, the stained cells were extracted with an extraction solution and then measured at 560 nm using an Omega Plate Reader (BMG Labtech, Ortenberg, Germany).

### 2.5. RPE Cell Collagen Gel Contraction

The RPE cell mechanical properties were determined using the CytoSelect 48-Well Cell Contraction Assay (Cell Biolabs, San Diego, CA, USA). All procedures were performed according to the manufacturer’s instructions. Briefly, RPE cells were resuspended in the medium at 2.0 × 10^4^ RPE cells/well. The RPE cell contraction matrix was prepared by mixing two parts of RPE cells and eight parts of the collagen gel working solution, and then 250 mL aliquots of the RPE cell collagen gel working solution were dispensed into a 48-well cell contraction plate. The RPE cell collagen gel working solution was polymerized for 1 h at 37 °C and 5% CO_2_ and treated with Nano-GO (20 and 40 mg/mL). Subsequently, the RPE cell contraction matrix was monitored for 12, 24, and 48 h at 37 °C and 5% CO_2_.

### 2.6. ROS Production in RPE Cells

Intracellular ROS levels were analyzed using 5- (and-6) -chloromethyl-2′,7′-dichlorodihydrofluorescein diacetate acetyl ester (CM-H2DCFDA; Thermo Fisher Scientific, Waltham, MA, USA). All procedures were performed according to the instructions set by the manufacturer. In brief, after Nano-GO treatment, the RPE cells were rinsed with PBS (phosphate-buffered saline) and incubated with CM-H2DCFDA for 30 min in the dark. Thereafter, endogenous ROS levels were measured based on the fluorescence unit using fluorescein isothiocyanate on a flow cytometer (Attune NxT Flow cytometer, Thermo Fisher Scientific, Pasadena, CA, USA). Hydrogen peroxide (H_2_O_2_, 500 μM), which is known to induce ROS production, was used as a positive control [27].

### 2.7. Immunofluorescence Microscopy

RPE cells were incubated with E-cadherin (1:200), N-cadherin (1:100), vimentin (1:200), and α-SMA (1:100) (Cell Signaling Technology, Beverly, MA, USA) overnight at 4 °C and stained with a secondary antibody conjugated to Alexa Fluor 488 (1:500) dilution and Texas Red-X (1:500) dilution (Thermo Fisher Scientific, Waltham, MA, USA) for 3 h in the dark. Subsequently, the RPE cells were stained with DAPI (4′,6-diamidino-2-phenylindole), which was coupled with nuclear acid staining to determine the cell status. RPE cells were mounted, and images were taken using a Carl Zeiss fluorescence microscope (Carl Zeiss, Oberkochen, Germany).

### 2.8. Western Blotting

RPE cells were harvested and lysed using M-PER (Mammalian Protein Extraction Reagent, Thermo Fisher Scientific, Waltham, MA, USA). All procedures were performed according to the instructions set by the manufacturer. All protein concentrations were determined using a Bio-Rad protein assay kit (Bio-Rad, Hercules, CA, USA) as per the instruction manual. The gel was separated using Mini-PROTEAN Precast Gels (Bio-Rad, CA, USA) and transferred onto a Hybond polyvinylidene difluoride membrane (Amersham Biosciences, Piscataway, NJ, USA). Immunodetection was performed using E-cadherin, N-cadherin, vimentin, α-SMA, MMP-2, MMP-9, and α-tubulin (Cell Signaling Technology, Beverly, MA, USA) in the SignalBoost Immunoreaction Enhancer Kit assay (Sigma–Aldrich, St. Louis, MO, USA). The protein bands were observed using an enhanced Pierce ECL Western Blotting Substrate (Thermo Fisher Scientific, Waltham, MA, USA) and quantified as the ratio of the target protein band intensity to the α-tubulin band intensity.

### 2.9. siRNA Transfection

RPE cells were transfected with human small interfering RNA (siRNA) against PLD1, PLD2, and control siRNA using the X-tremeGENE siRNA Transfection Reagent (Sigma–Aldrich, St. Louis, MO, USA) according to the instructions set by the manufacturer. The total amount of siRNA used for each well was normalized to that of the scrambled RNA (control siRNA).

### 2.10. PLD Enzyme Activity in RPE Cells

PLD enzyme activity was detected using the Amplex Red Phospholipase D Assay Kit (Thermo Fisher Scientific, Waltham, MA, USA). All procedures were performed according to the manufacturer’s instructions. In brief, the RPE cells were washed with PBS and extracted using three freeze-thaw cycles in a lysis solution. Samples were mixed with the Amplex Red reaction buffer and PLD enzymatic activity was detected using the FLUOstar Omega Multi-Mode Microplate Reader (BMG Labtech).

### 2.11. Statistical Analysis

All assays were independently repeated at least three times. All statistical parameters are presented as the mean ± standard error of the mean (SEM). Statistical analyses were performed using one-way analysis of variance (ANOVA) followed by Dunn’s *post-hoc* test. A value of *p* < 0.01 or *p* < 0.05 was considered significant.

## 3. Results

### 3.1. Zetasizer, XRD, and FTIR Analyses of Nano-GO

Nano-GO was characterized using a Zetasizer, X-ray diffractometer, and FTIR spectrometer, as previously reported. Zetasizer analysis was performed to study the size distribution and zeta potential of Nano-GO to confirm the optimal hydrodynamic diameter and effective surface charge that improve the stability of colloidal dispersions. Our results showed that the optimal hydrodynamic diameter was 87.8 ± 0.7 nm (Figure 1A), and the optimal zeta potential was −33.7 ± 1.2 mV with a single peak (Figure 1B), which indicates stable dispersion via interparticle electrostatic repulsion. The XRD pattern of the Nano-GO showed characteristic peaks that were consistent with the formation of oxide groups, such as epoxide, carbonyl, and hydroxide groups. Based on the XRD pattern, peaks were observed at 10.07°, which are indicative of the (001) planes (Figure 1C). As reported earlier, these peaks indicate the chemical oxidation of graphite to GO [29]. Additionally, the functional group analysis of Nano-GO using an FTIR spectrophotometer characterized an alkoxy C–O stretch at 1031 cm^−1^, epoxy C–O stretch at 1227 cm^−1^, aromatic C = C stretch at 1608 cm^−1^, carboxyl C = O stretch at 1719 cm^−1^, C–H stretch at 2893 cm^−1^, and hydroxy–OH groups at 3359 cm^−1^ (Figure 1D). The TEM images of Nano-GO are presented in Figure 1E.

### 3.2. Nano-GO Promoted RPE Cell Migration

The Nano-GO characterization was consistent with that obtained in a previous study. The human RPE cells ARPE-19 have generally been used to evaluate RPE studies [30]. To investigate whether Nano-GO exhibited EMT progression in RPE cells, we first assessed the cytotoxic effects of Nano-GO treatment using the Cell Counting Kit-8 assay. As shown in Figure 2A, the cell viability was slightly reduced to 81.3 ± 0.6% at 80 μg/mL for 24 h, and this level of reduced cell viability gradually increased with time and concentration, up to 200 μg/mL. At 200 μg/mL Nano-GO for 48 and 72 h, the cell viability was inhibited to approximately 80%. However, Nano-GO treatment did not exert cytotoxicity at 20 and 40 μg/mL for 24 h, 48 h, and 72 h. Consequently, the RPE cells were treated with Nano-GO at 20 and 40 μg/mL for 24 h and 48 h in the subsequent experiments. Recent evidence suggests that the EMT process of RPE cells is an important step in numerous intraocular fibrotic disorders, including proliferative vitreoretinopathy and age-related macular degeneration. In this EMT process, RPE cells lose epithelial characteristics through cell migration ability, production of extracellular matrix, and expression of mesenchymal markers [21,28,31]. The RPE cell migration assay was performed to evaluate the ability of Nano-GO to induce RPE cell motility. The RPE cell migration showed that the straight-line gap was repopulated by RPE cells. The straight-line gap of the control group was slightly repopulated by the RPE cells, while a remarkable straight-line gap closure was observed in the Nano-GO group (20 and 40 μg/mL) with more rapid gap coverage by 24 h and complete coverage by 48 h (Figure 2B). We found that RPE cell motility progressively increased following treatment. We further investigated RPE cell migration during Nano-GO treatment for 24 h. After treating RPE cells with 20 and 40 μg/mL of Nano-GO, the transwell assay showed that the RPE cells which moved to the lower membrane of the chamber were significantly and dose-dependently greater with the Nano-GO treatment than in the control (Figure 2C). The migratory RPE cells were extracted with extraction solution and then quantified at the optical density OD560. We found that Nano-GO (20 and 40 μg/mL) significantly increased the cell migration to 140.1% and 226.8%, respectively (Figure 2D), suggesting that Nano-GO induced the migratory capability of RPE cells. This result suggests that Nano-GO treatment resulted in high RPE cell mobility.

### 3.3. Nano-GO Induced RPE Cell-Mediated Collagen Gel Contraction

Collagen gel contraction through RPE cells is a hallmark of degenerative retinal diseases and plays a critical role in the development of the EMT process [24]. To determine whether Nano-GO treatment could affect the RPE cell-mediated collagen gel contraction, Nano-GO (20 and 40 μg/mL) was added to the 3-dimensional collagen gel and observed at 12 h, 24 h, and 48 h. Figure 3A shows that treatment of RPE cells with Nano-GO (20 and 40 μg/mL) resulted in a marked contraction of collagen gel. The areas of collagen gel were significantly reduced to 76.1% and 36.8% of the initial area after 48 h, respectively (Figure 3B). These collagen gel contractions were clearly driven by the Nano-GO-treated RPE cells because the gels without Nano-GO retained a larger area. This gel-contracting effect of Nano-GO was similar to that observed in RPE cell migration.

### 3.4. Nano-GO Induced Intracellular ROS Production in RPE Cells

Since ROS production leads to severe EMT, intracellular ROS levels are considered to be potential mediators of degenerative retinal diseases [27,28]. Several studies have suggested that Nano-GO induces endogenous ROS production [7,8,32]. We sought to confirm whether endogenous ROS levels were increased in Nano-GO-treated RPE cells. To do this, we analyzed the intracellular fluorescence intensity of the probe CM-H2DCFDA using fluorescence microscopy and flow cytometry. The fluorescence microscopy results indicated that the CM-H2DCFDA staining images displayed slight staining in the control RPE cells, which were markedly stained in Nano-GO (20 and 40 μg/mL; Figure 4A). The flow cytometry results showed that the intracellular fluorescence intensity was 9.8% in the control, whereas the groups treated with 20 and 40 μg/mL Nano-GO had an intensity of 51.0% and 86.2%, respectively. Hydrogen peroxide (H_2_O_2_), a generator of ROS production, was added to the positive control. We used 500 μM H_2_O_2_, which was verified without affecting the cell viability, as previously reported. The positive control also significantly increased the intracellular fluorescence intensity in the flow cytometry assay. Our data showed that the Nano-GO treatment significantly increased intracellular ROS production compared to the control (Figure 4B).

### 3.5. Immunofluorescence Staining and Western Blot Analysis of Nano-GO in RPE Cells

Numerous organic molecules have been found to be involved in regulating the RPE EMT process. In particular, E-cadherin, as an intercellular adhesion molecule, can attenuate the EMT process, while N-cadherin, α-SMA, and vimentin can promote EMT as mesenchymal molecules [25,26]. The protein expression levels and subcellular localization of E-cadherin, N-cadherin, vimentin, and α-SMA were determined through immunofluorescence microscopy. A normal protein expression of E-cadherin, an epithelial marker, was detected in control RPE cells, but this protein expression was significantly reduced through treatment with Nano-GO (Figure 5A). Furthermore, Nano-GO at 40 μg/mL significantly induced the protein expression of N-cadherin (Figure 5B), α-SMA (Figure 5C), and vimentin (Figure 5D) as mesenchymal markers. To further investigate the protein expression of E-cadherin, N-cadherin, vimentin, and α-SMA in RPE cells, these cells were treated with Nano-GO (20 and 40 μg/mL). Compared with the control, the RPE cells treated with Nano-GO showed a significantly suppressed E-cadherin protein expression. In contrast, the protein levels of N-cadherin, vimentin, and α-SMA were upregulated after treatment with Nano-GO (Figure 5E). Concurrently, matrix metalloproteinases (MMP-2 and MMP-9), a family of zinc-dependent endopeptidases, can also break down the extracellular matrix to facilitate the EMT process. Several studies have shown that an induced matrix metalloproteinase expression is associated with EMT progression. Therefore, the ability of Nano-GO to upregulate the protein expression of MMP-2 and MMP-9 in RPE cells was investigated. Nano-GO at 40 μg/mL led to the induction of the protein expression of MMP-2 and MMP-9 in RPE cells (Figure 5E). Together, these results suggest that Nano-GO facilitates the RPE EMT process via the regulation of EMT-related molecules.

### 3.6. Nano-GO Mediated RPE EMT Process Associated with PLD

Phosphatidic acid (PA) is a bioactive signaling lipid and cell membrane component that plays a pivotal role in many physiological processes, including cell growth, migration, survival, transformation, membrane trafficking, cytoskeletal reorganization, and differentiation. PLD is the only enzyme that hydrolyzes phosphatidylcholine (PC) to PA and choline-free polar head groups [33,34,35]. PLD is expressed in many different cell types, including macrophage, colon, breast, gastric, and RPE cells. Recently, interesting relationships have been established between PLD and RPE cells. Dysregulated PLD has been suggested to play a key role in RPE EMT progression in several forms of retinal degeneration [36,37,38]. Therefore, we determined whether the Nano-GO-mediated EMT process is associated with PLD and whether it is mediated by PLD1 and PLD2. An siRNA system was used to reduce the activation of PLD1 and PLD2. Compared with the si-control group treated with Nano-GO (40 μg/mL), the RPE cells transfected with small interfering phospholipase D1 or D2 (si-PLD1 or si-PLD2) had significantly downregulated PLD enzymatic activity (Figure 6A). Subsequently, we performed RPE cell migration assays to clarify whether the Nano-GO-induced PLD enzymatic activation was caused by RPE cell migration in Nano-GO-treated RPE cells. We measured the migratory properties of RPE cells treated with 40 μg/mL of Nano-GO and compared the results with those of RPE cells transfected with si-PLD1 or si-PLD2. Interestingly, compared to treatment with the si-control group treated with Nano-GO (40 μg/mL), transfection with si-PLD1 or si-PLD2 significantly reversed the Nano-GO-induced migratory capabilities of RPE cells (Figure 6B). To elucidate the involvement of PLD signaling in intracellular ROS generation in Nano-GO-treated RPE cells, siRNA against PLD1 and PLD2 was used. The knockdown of PLD1 and PLD2 blocked the Nano-GO-induced ROS production (Figure 6C). Therefore, our results strongly suggest that Nano-GO-mediated RPE cell migration and ROS production are highly dependent on PLD signaling.

## 4. Discussion

The retina is constantly exposed to detrimental nanomaterials and pathogens. Recent studies have demonstrated that Nano-GO might modulate the EMT process in lung, breast, prostate, and liver cancer [7,17]. To the best of our knowledge, no data regarding the influence of Nano-GO on the RPE EMT process are currently available. Due to the limited availability of primary cells, the immortalized human retinal epithelial cell line ARPE-19 was used to further explore and expand on existing research related to the RPE EMT process. ARPE-19 cells retain many of the functions and morphology of normal human retinal epithelial cells and are proposed to be valuable in vitro models for studying degenerative retinal disease, and for investigating the consequences of exposure to detrimental nanomaterials. Regarding cytotoxicity, our results showed that only higher concentrations and longer exposure times of Nano-GO significantly decreased cell viability. However, Nano-GO (20 and 40 μg/mL for 24 h, 48 h, and 72 h) did not significantly lower the cytotoxicity of human RPE cells. Other studies applying the same or similar Nano-GO concentrations on human RPE cells obtained equivalent results, and proliferative effects were even sometimes observed, while higher Nano-GO doses were cytotoxic. We used these concentrations to assess whether Nano-GO could affect RPE cell migration as an EMT process. We found that Nano-GO stimulated the RPE cell migration ability, depending on the concentration, especially at a concentration of 40 µg/mL.

Nano-GO has been reported to upregulate oxidative stress in macrophages, lymphocytes, and embryonic fibroblast cells [39,40]. Normal ROS production is essential for maintaining normal physiological functions in human RPE cells. However, ROS production abnormally increases in response to the RPE EMT process. Our results similarly show that Nano-GO can modulate intracellular ROS production in human RPE cells, especially in the RPE EMT process. Our studies have shown that Nano-GO modulate EMT-related biomarkers (E-cadherin, N-cadherin, vimentin, and α-SMA) in human RPE cells. E-cadherin is essential for the maintenance of epithelial cells in human RPE cells, and its downregulation is related to EMT in human RPE cells. Our results also indicate that downregulation of E-cadherin by Nano-GO can induce EMT and may lead to increased human RPE cell migration. N-cadherin, vimentin, and α-SMA act as mesenchymal molecular patterns leading to the RPE EMT process.

PLD is involved in many biological processes, including cell proliferation, differentiation, and migration [33,34,35]. Numerous studies have shown that PLD plays a role in various pathophysiological processes, such as inflammation, metabolic syndrome, nonalcoholic fatty liver disease, and Alzheimer’s disease. At the same time, PLD is also one of the key molecules involved in the EMT process in other tissues [36,37,38]. Our research confirms the value and function of PLD in an RPE cells. Compared with previous studies, our study was further validated by identifying possible signaling pathways and exploring significant differences. Nano-GO has been demonstrated to be involved in the regulation of several physiological and pathological processes, such as tumor growth, progression, and metastasis. We also found that human RPE cells had PLD gene expressions and PLD enzymatic activity that were further induced by Nano-GO treatment. The EMT process of Nano-GO in human RPE cells appears to be associated with PLD signals. Nano-GO was found to affect PLD enzymatic activity the knockdown of PLD1 and PLD2 suppressed migration compared to Nano-GO-treated RPE cells, as well as reducing the intracellular ROS generation when compared to Nano-GO on its own. These PLD1 and PLD2 knockdown patterns might explain the human RPE EMT process when considered alongside the study showing that the human RPE EMT process with Nano-GO seems to signal by means of PLD signals. It should be emphasized that activation of the PLD signal in an RPE cell guarantees participation in response to the RPE EMT process. The crosstalk between the EMT process initiated by PLD signal activation is still unknown, and this might be interesting to explore in future studies. Notably, we found that the Nano-GO-mediated RPE EMT process was associated with the enhancement of PLD activation.

## 5. Conclusions

Our study was designed to test our hypothesis that Nano-GO promotes EMT by regulating PLD signaling in RPE cells. Human RPE cells undergo EMT-like pathological changes, leading to conditions that can lead to degenerative retinal diseases. In Nano-GO-treated RPE cells, we found that RPE cell migration was significantly upregulated. Using the cell contraction assay, we also clarified the effect of Nano-GO-treated RPE cells on collagen gel contraction. Similar results were observed where Nano-GO alleviated the intracellular ROS production. Moreover, western blotting results showed that Nano-GO downregulated E-cadherin and upregulated N-cadherin, vimentin, α-SMA, MMP-2, and MMP-9 in treated RPE cells. To determine whether Nano-GO contributes to the EMT process, a persistently knocked-down PLD system was used. The results showed that the knockdown of PLD1 and PLD2 significantly reversed the EMT process of Nano-GO. These findings contribute toward a deeper understanding of degenerative retinal diseases and the pathogenic role of Nano-GO.

## Figures and Tables

**Figure 1 nanomaterials-11-02546-f001:**
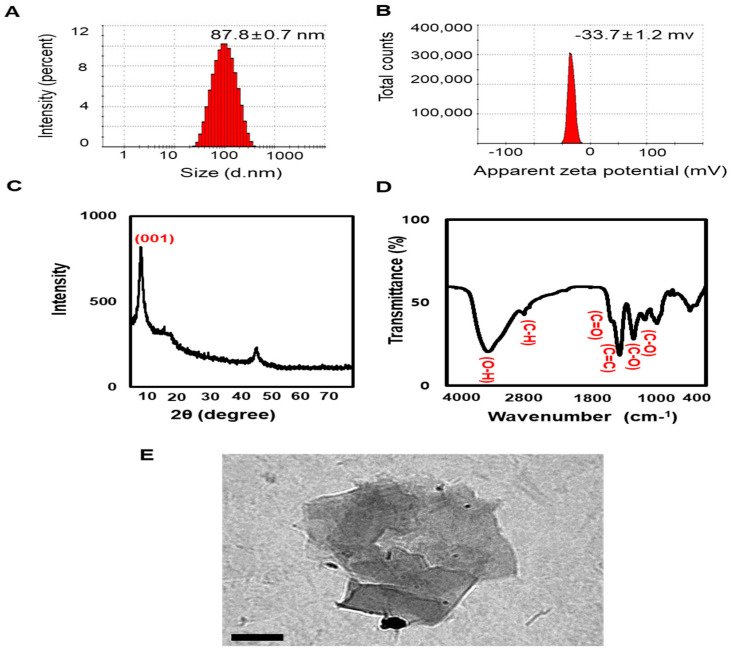
Physicochemical characterization of Nano-GO. Zetasizer showing the size distribution (**A**) and zeta potential (**B**) of Nano-GO. (**C**) Representative XRD pattern of Nano-GO. (**D**) FTIR spectra of Nano-GO. (**E**) TEM images of Nano-GO (scale bar; 200 nm).

**Figure 2 nanomaterials-11-02546-f002:**
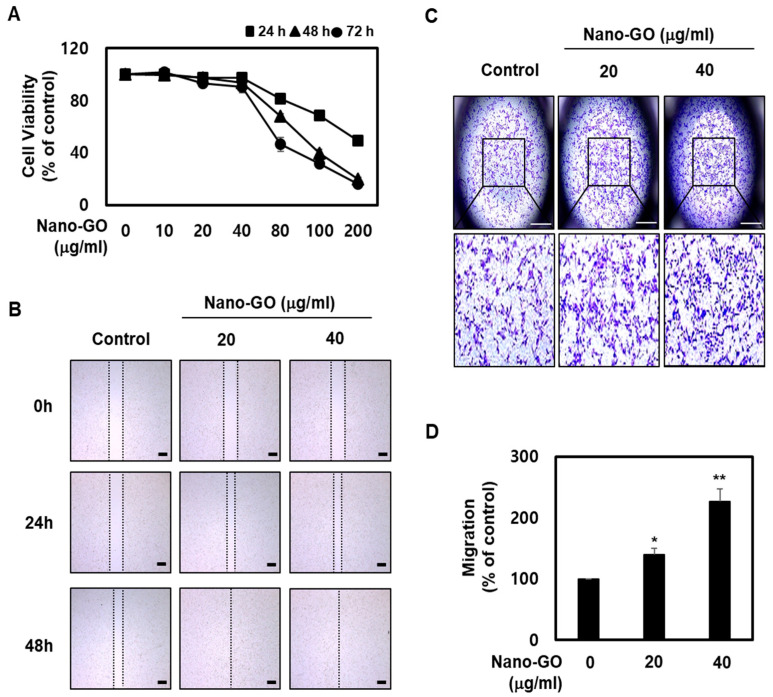
Effect of Nano-GO on RPE cell migration. (**A**) RPE cells were treated with Nano-GO (0, 10, 20, 40, 80, 100, and 200 μg/mL) for 24, 48, and 72 h. RPE cell viability was measured using the Cell Counting Kit-8 assay. (**B**) RPE cells were treated with Nano-GO (20 and 40 μg/mL) for 24 or 48 h. Migratory RPE cell images were taken at 0, 24, and 48 h after scratching was applied. The lines show the boundaries of the transferred RPE cells (scale bar; 100 μm). Transwell migration assay. RPE cells were allowed to move through the membrane for 24 h in the presence or absence of Nano-GO (20 and 40 μg/mL). Representative images of migrated RPE cells (**C**) and quantification (**D**) are shown (scale bar; 50 μm). All data are presented as the mean ± SEM (*n* = 3). * *p* < 0.05 and ** *p* < 0.01 compared to the Control.

**Figure 3 nanomaterials-11-02546-f003:**
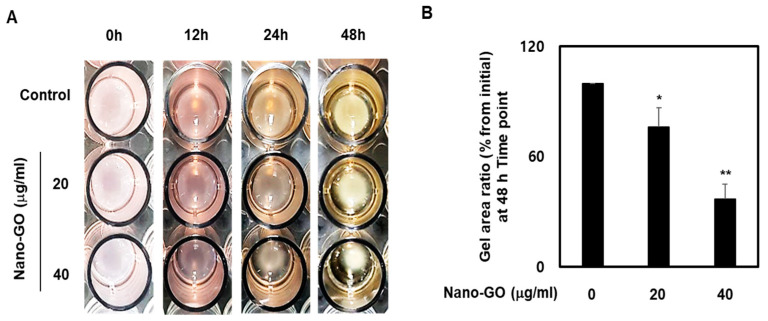
Effect of Nano-GO on RPE cell collagen gel contraction. The RPE cells were treated with Nano-GO (20 and 40 μg/mL) for 48 h. Representative photo (**A**) and quantitative analysis (**B**) show the action of Nano-GO-treated or untreated RPE cells on collagen gel contraction. The gel area of each collagen gel was measured at 48 h. All data are presented as the mean ± SEM (*n* = 3). * *p* < 0.05 and ** *p* < 0.01 compared to the Control.

**Figure 4 nanomaterials-11-02546-f004:**
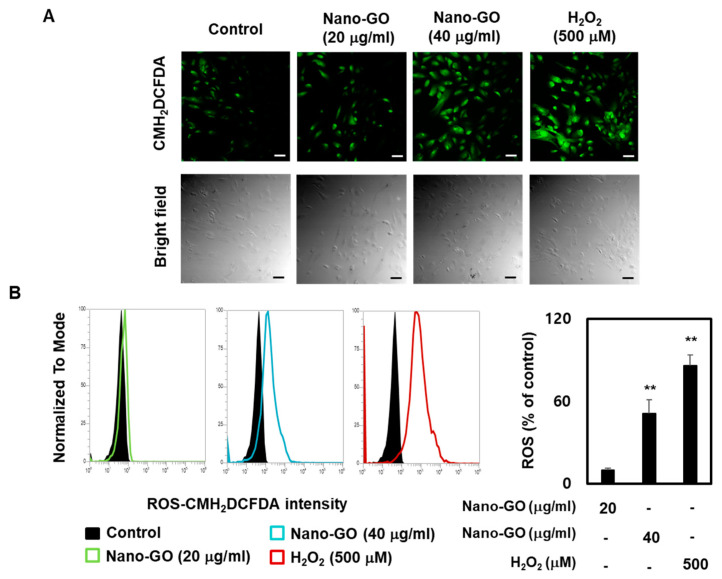
Effect of Nano-GO on intracellular ROS production in RPE cells. The RPE cells were treated with Nano-GO (20 and 40 μg/mL) for 24 h. (**A**) Intracellular ROS production was visualized by Carl Zeiss fluorescence microscopy using CM-H_2_DCFDA staining (scale bar; 50 μm). (**B**) Representative flow cytometer-based image (upper panel) and quantitative analysis (lower panel) indicates fluorescence intensity using CM-H_2_DCFDA staining. All data are presented as the mean ± SEM (*n* = 3). * *p* < 0.05 and ** *p* < 0.01 compared to the Control.

**Figure 5 nanomaterials-11-02546-f005:**
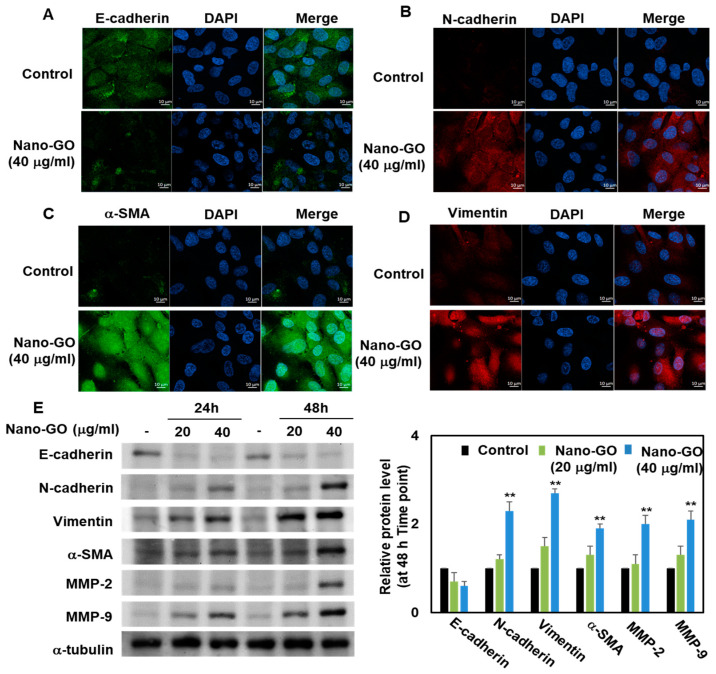
Effect of Nano-GO on protein expression of E-cadherin, N-cadherin, vimentin, α-SMA, MMP-2, and MMP-9. RPE cells were immunostained for E-cadherin (**A**), N-cadherin (**B**), α-SMA (**C**), and vimentin (**D**) in the absence or presence of Nano-GO (20 and 40 μg/mL, for 24 h). Immunofluorescence images were taken using a 63× oil objective (scale bar; 10 μm). (**E**) Representative western blotting-based image (right panel) and quantitative analysis (left panel) show the protein expression of E-cadherin, N-cadherin, vimentin, α-SMA, MMP-2, and MMP-9 in RPE cells. DAPI was used to determine the nuclei. All data are presented as the mean ± SEM (*n* = 3). * *p* < 0.05 and ** *p* < 0.01 compared to the Control.

**Figure 6 nanomaterials-11-02546-f006:**
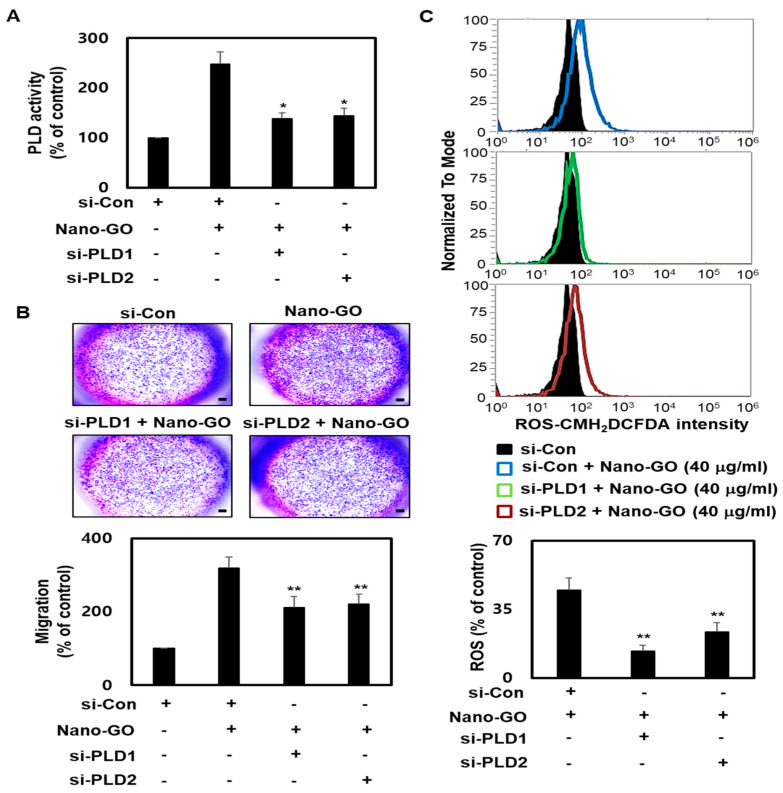
Effect of Nano-GO on RPE cell migration and ROS production through the PLD signal. RPE cells were treated with si-Control, si-PLD1, or si-PLD2, and then treated with Nano-GO (40 μg/mL) for 24 h. (**A**) PLD enzymatic activity was assessed using the Amplex Red Phospholipase D Assay Kit (**B**) RPE cell migration was assessed via cell migration assay (scale bar; 50 μm). (**C**) Intracellular ROS production was assessed via CM-H_2_DCFDA using flow cytometry. All data are presented as the mean ± SEM (*n* = 3). * *p* < 0.05 and ** *p* < 0.01 compared to the Control.

## Data Availability

The data that support the findings of this study are available from the corresponding author upon reasonable request.

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
