# Peer review of "Nano-Graphene Oxide-Promoted Epithelial–Mesenchymal Transition of Human Retinal Pigment Epithelial Cells through Regulation of Phospholipase D Signaling"

_nanomaterials, 2021, doi:10.3390/nano11102546_

Round 1

Reviewer 1 Report

This paper have demonstrated that Nano-GO promotes RPE cell migration through PLD-mediated ROS production. The study is interesting and meaningful. It can be accepted for publication after addressing following issues.

  1. Some sentences are ambiguous. For example, in the introduction section, the sentence “Recently, Nano-GO has shown great potential as a biocompatible scaffold for drug delivery. However, there remain certain problems for human health”, and “Because Nano-GO is widely applied, it has been proven to be involved in the regulation of several physiological and pathological processes, such as tumor growth, progression, and metastasis”. The causal relationship between these two is farfetched. The language should be polished.
  2. In Page 5, the expression for figure 1(E), the TEM of Nano-GO is not clear. As for figure 2(A), the abscissa and ordinate do not correspond to the point. The font size of texts in several figures are not appropriate, which may prevent the readers from getting valid information. It is necessary to make some improvements on that.
  3. There are inconsistencies between pictures and texts in the article. Figure 2(D), Abscissa: the concentration of Nano-GO is 10 and 20. However, the descriptions and illustrations of figure 2(D) are 20 and 40. The similar problem appears in Figure 3(B) . The abscissa of the second column in Figure 4(B) is ambiguous. The concentration of Nano-GO shown is 20+40 μg/mL. Moreover, the existence of si-Control in the cylindrical comparison diagram of Figure 6(B) and (C) is in conflict. Please check carefully.
  4. Due to the imaging of these cell is ambiguous in figure 4(A), the cell shape of Nano-GO (20 μg/mL) is a little different from that of Nano-GO (40 μg/mL), images of bright field should be provided.
  5. This paper describes the phenomenon detaliedly, while the mechanism is not well explored. I think the author should properly explain the mechanism of Nano-GO promoting EMT in RPE cells.

Author Response

Thank you very much for allowing us to revise our manuscript entitled, “Nano-graphene oxide-promoted epithelial–mesenchymal transition of human retinal pigment epithelial cells through regulation of phospholipase D signaling (nanomaterials-1351225)”. We appreciate the reviewers for their constructive comments, which were very helpful for improving our paper. The manuscript has been carefully revised according to the reviewers’ comments. The revisions are marked in red in the revised manuscript. The detailed responses (in BOLD type) to the comments are provided below.

This paper have demonstrated that Nano-GO promotes RPE cell migration through PLD-mediated ROS production. The study is interesting and meaningful. It can be accepted for publication after addressing following issues.

  1. Some sentences are ambiguous. For example, in the introduction section, the sentence “Recently, Nano-GO has shown great potential as a biocompatible scaffold for drug delivery. However, there remain certain problems for human health”, and “Because Nano-GO is widely applied, it has been proven to be involved in the regulation of several physiological and pathological processes, such as tumor growth, progression, and metastasis”. The causal relationship between these two is farfetched. The language should be polished.

Response: This sentence has been revised according to the reviewer's comments.

[Since the introduction of graphene-based nanomaterials in a two-dimensional carbon sheet in 2004, various applications of graphene oxide (GO) have been anticipated in many scientific fields, such as sensing, electronics, optical energy, biomedicine, and biotechnology.[1-3] Nano-graphene oxide (Nano-GO) is a granular material with sizes ranging from 20 to 100 nm. With its unique properties, Nano-GO has revolutionized biomedical applications through targeted drug delivery and disease diagnosis [4,5]. However, Nano-GO has become a topic of growing concern for human health risks [6, 7]. Similar to other particulate matter, several studies have shown that GO is harmful to human health.[8-10] Some studies have reported that GO induces reactive oxygen species (ROS), which are capable of causing plasma membrane damage, mitochondrial injury, immune responses, and programmed cell death.[11-13] It has been reported that GO may lead to pro-inflammation, cell death, thrombus formation, and cancer metastasis, which appears to be induced by increased ROS production.[14-16] Furthermore, GO has been demonstrated to exert advantageous and disadvantageous effects on tumor progression. As a potential tumor promoter, GO can promote the metastasis of human lung, breast, prostate, and liver cancer.[17,18]]

  1. In Page 5, the expression for figure 1(E), the TEM of Nano-GO is not clear. As for figure 2(A), the abscissa and ordinate do not correspond to the point. The font size of texts in several figures are not appropriate, which may prevent the readers from getting valid information. It is necessary to make some improvements on that.

Response: We apologize for the oversight and lack of clarity. We have modified in Figure 1(E) and 2(A).

  1. There are inconsistencies between pictures and texts in the article. Figure 2(D), Abscissa: the concentration of Nano-GO is 10 and 20. However, the descriptions and illustrations of figure 2(D) are 20 and 40. The similar problem appears in Figure 3(B) . The abscissa of the second column in Figure 4(B) is ambiguous. The concentration of Nano-GO shown is 20+40 μg/mL. Moreover, the existence of si-Control in the cylindrical comparison diagram of Figure 6(B) and (C) is in conflict. Please check carefully.

Response: We apologize for the oversight and lack of clarity. We have modified in Figure 2(D) 3 (B), 4(B) and 6 (B, C).

  1. Due to the imaging of these cell is ambiguous in figure 4(A), the cell shape of Nano-GO (20 μg/mL) is a little different from that of Nano-GO (40 μg/mL), images of bright field should be provided.

Response: According to the comments of the reviewer, we have modified figure 4 (A).

  1. This paper describes the phenomenon detaliedly, while the mechanism is not well explored. I think the author should properly explain the mechanism of Nano-GO promoting EMT in RPE cells.

Response: Thank you very much for your comments, we have added more detailed information in Discussion.

[PLD is involved in many biological processes, including cell proliferation, differenti-ation, and migration [33-35]. Numerous studies have shown that PLD plays a role in var-ious pathophysiological processes such as inflammation, metabolic syndrome, nonalco-holic fatty liver disease, and Alzheimer's disease. At the same time, PLD is also one of the key molecules involved in the EMT process in other tissues [36-38]. Our research confirms the value and function of PLD in an RPE cells. Compared with previous studies, our study was further validated by identifying possible signaling pathways and exploring signifi-cant differences. Nano-GO has been demonstrated to be involved in the regulation of sev-eral physiological and pathological processes such as tumor growth, progression and metastasis. We also found that human RPE cells had PLD gene expressions and PLD en-zymatic activity that were further induced by Nano-GO treatment. The EMT process of Nano-GO in human RPE cells appears to be associated with PLD signals. Nano-GO was found to affect PLD enzymatic activity the knockdown of PLD1 and PLD2 suppressed migration compared to Nano-GO-treated RPE cells, as well as reducing the intracellular ROS generation when compared to Nano-GO on its own. These PLD1 and PLD2 knock-down patterns might explain the human RPE EMT process when considered alongside the study showing that the human RPE EMT process with Nano-GO seems to signal by means of PLD signals. It should be emphasized that activation of the PLD signal in an RPE cell guarantees participation in response to the RPE EMT process. The crosstalk be-tween the EMT process initiated by PLD signal activation is still unknown, and this might be interesting to explore in future studies. Notably, we found that the Nano-GO-mediated RPE EMT process was associated with the enhancement of PLD activation.]

Reviewer 2 Report

The manuscript presented by Park and collaborators is focused on the potential toxic effect of Nano-GO, a material widely used in various biotechnological approaches and of which it is known the relationship with the onset, progression and metastatic capacity of various cancers, on the RPE and the possible induction of the EMT process. The EMT process is often correlated with dysfunction of the RPE which then leads to the development of retinal degenerative diseases such as AMD. Although the study is interesting to be able to deepen the knowledge about biomaterials and their potential long-term toxic effects the manuscript needs some corrections. 
In general I suggest to the authors to reread the manuscript carefully and to change some terminology, for example in row 388: I would replace the term "regulates" with "modulate".
Figure 4: it would be appropriate to add to the panel of images of immunocytochemistry also the image related to the positive control, that is the cells stressed with H2O2; moreover, always in relation to Figure 4, the authors, at line 310, write: "Compared with the control, the RPE cells 
 treated with Nano-GO showed a significantly suppressed N-cadherin protein expression..." I believe that N-cadherin is to be replaced with E-cadherin.
Finally, I believe that the most critical part is the one related to the role of PLD; at the present time (line 396-409) the authors have only summarized the data already mentioned in the results without hypothesize a mechanism of action with which we can explain the crosstalk Nano-GO-RPE EMT-PLD, I would advise the authors to deepen this part possibly with molecular biology experiments to evaluate a possible pathway activated by PLD in the toxicity trigger by Nano-GO.

Author Response

Thank you very much for allowing us to revise our manuscript entitled, “Nano-graphene oxide-promoted epithelial–mesenchymal transition of human retinal pigment epithelial cells through regulation of phospholipase D signaling (nanomaterials-1351225)”. We appreciate the reviewers for their constructive comments, which were very helpful for improving our paper. The manuscript has been carefully revised according to the reviewers’ comments. The revisions are marked in red in the revised manuscript. The detailed responses (in BOLD type) to the comments are provided below.

The manuscript presented by Park and collaborators is focused on the potential toxic effect of Nano-GO, a material widely used in various biotechnological approaches and of which it is known the relationship with the onset, progression and metastatic capacity of various cancers, on the RPE and the possible induction of the EMT process. The EMT process is often correlated with dysfunction of the RPE which then leads to the development of retinal degenerative diseases such as AMD. Although the study is interesting to be able to deepen the knowledge about biomaterials and their potential long-term toxic effects the manuscript needs some corrections.

In general I suggest to the authors to reread the manuscript carefully and to change some terminology, for example in row 388: I would replace the term "regulates" with "modulate".

Response: Thank you very much for your comments; I have corrected this.

Figure 4: it would be appropriate to add to the panel of images of immunocytochemistry also the image related to the positive control, that is the cells stressed with H2O2; moreover, always in relation to Figure 4, the authors, at line 310, write: "Compared with the control, the RPE cells

 treated with Nano-GO showed a significantly suppressed N-cadherin protein expression..." I believe that N-cadherin is to be replaced with E-cadherin.

Response: We apologize for the oversight and lack of clarity. We have modified in Figure 4 (A).

Response: Thank you very much for your comments; I have corrected this (E-cadherin).

Finally, I believe that the most critical part is the one related to the role of PLD; at the present time (line 396-409) the authors have only summarized the data already mentioned in the results without hypothesize a mechanism of action with which we can explain the crosstalk Nano-GO-RPE EMT-PLD, I would advise the authors to deepen this part possibly with molecular biology experiments to evaluate a possible pathway activated by PLD in the toxicity trigger by Nano-GO.

Response: Thank you very much for your comments, we have added more detailed information in Discussion.

[PLD is involved in many biological processes, including cell proliferation, differenti-ation, and migration [33-35]. Numerous studies have shown that PLD plays a role in var-ious pathophysiological processes such as inflammation, metabolic syndrome, nonalco-holic fatty liver disease, and Alzheimer's disease. At the same time, PLD is also one of the key molecules involved in the EMT process in other tissues [36-38]. Our research confirms the value and function of PLD in an RPE cells. Compared with previous studies, our study was further validated by identifying possible signaling pathways and exploring signifi-cant differences. Nano-GO has been demonstrated to be involved in the regulation of sev-eral physiological and pathological processes such as tumor growth, progression and metastasis. We also found that human RPE cells had PLD gene expressions and PLD en-zymatic activity that were further induced by Nano-GO treatment. The EMT process of Nano-GO in human RPE cells appears to be associated with PLD signals. Nano-GO was found to affect PLD enzymatic activity the knockdown of PLD1 and PLD2 suppressed migration compared to Nano-GO-treated RPE cells, as well as reducing the intracellular ROS generation when compared to Nano-GO on its own. These PLD1 and PLD2 knock-down patterns might explain the human RPE EMT process when considered alongside the study showing that the human RPE EMT process with Nano-GO seems to signal by means of PLD signals. It should be emphasized that activation of the PLD signal in an RPE cell guarantees participation in response to the RPE EMT process. The crosstalk be-tween the EMT process initiated by PLD signal activation is still unknown, and this might be interesting to explore in future studies. Notably, we found that the Nano-GO-mediated RPE EMT process was associated with the enhancement of PLD activation.]

therefore, further studies are needed. We will do our best to proceed with further studies on this topic.

Round 2

Reviewer 2 Report

Accepted in this revised form